# Pulse Oximetry Based on Quadrature Multiplexing of the Amplitude Modulated Photoplethysmographic Signals

**DOI:** 10.3390/s23136106

**Published:** 2023-07-02

**Authors:** Jeerasuda Koseeyaporn, Paramote Wardkein, Ananta Sinchai, Pattana Kainan, Panwit Tuwanut

**Affiliations:** 1School of Engineering, King Mongkut’s Institute of Technology Ladkrabang, Bangkok 10520, Thailand; jeerasuda.ko@kmitl.ac.th (J.K.); paramote.wa@kmitl.ac.th (P.W.); 60601155@kmitl.ac.th (P.K.); 2College of Advanced Manufacturing Innovation, King Mongkut’s Institute of Technology Ladkrabang, Bangkok 10520, Thailand; 3School of Information Technology, King Mongkut’s Institute of Technology Ladkrabang, Bangkok 10520, Thailand; panwit@it.kmitl.ac.th

**Keywords:** pulse oximetry, amplitude modulation, frequency-division multiplexing, synchronous detector

## Abstract

In this research, a pulse oximeter based on quadrature multiplexing of AM-PPG signals is proposed. The oximeter is operated by a microcontroller and employs a simple amplitude modulation technique to mitigate noise interference during SpO_2_ measurement. The two AM-PPG signals (RED and IR) are quadrature multiplexed using carrier signals with equal frequencies but a 90-degree phase difference. The study focused on noise interference caused by light intensity and hand movement. The experiment was conducted under three different levels of light intensity: 200 Lux, 950 Lux, and 2200 Lux. For each light intensity level, the SpO_2_ level was measured under three scenarios: hand still, shadow movement over the hand, and hand shaking. A comparison between the proposed technique and the conventional method reveals that the proposed technique offers a superior performance. The relative error of the measured SpO_2_ level using the proposed technique was less than 3.1% overall. Based on the study, the proposed technique is less affected by noise interference caused by light intensity and hand movement compared to the conventional method. In addition, the proposed technique has an advantage over contemporary methods in terms of computational complexity. Consequently, the proposed technique can be applied to wearable devices that include SpO_2_ measurement functionality.

## 1. Introduction

Due to the COVID-19 pandemic, a significant number of infections and deaths have occurred worldwide. Since the outbreak began in late 2019, the SARS-CoV-2 virus, responsible for COVID-19, has undergone changes over time. These mutations give rise to variants of the virus, some of which exhibit altered properties and cause different symptoms. Nevertheless, difficulty breathing, or shortness of breath, remains a common and severe symptom across various variants. Consequently, the use of a pulse oximeter, a device that monitors blood oxygen saturation in COVID-19 patients, is crucial. Currently, non-invasive pulse oximeters are widely used. In comparison to invasive types, non-invasive pulse oximeters do not require a blood sample, inflict no pain on the patient, eliminate the need for clinical laboratories, and provide diagnosis results quickly.

In commercial pulse oximeters, red and infrared (IR) light frequencies are utilized to penetrate the fingertip, with a single light detector measuring the absorbance levels of each light source. As a result, two signals known as photoplethysmography (PPG) signals are generated: one for the red light and another for the IR light. These signals are employed to calculate the ratio of light absorbance and estimate the level of blood oxygen saturation (SpO_2_).

To obtain an accurate measurement of SpO_2_, it is essential for the subject to maintain a still posture. Movement during the measurement process can introduce motion artifacts (MA) into the resulting PPG signals, leading to potential inaccuracies in the measured SpO_2_ levels. The movement problem is not a concern for measuring SpO_2_ in normal people, but it is critical for unconscious or Parkinson’s patients. Numerous studies have been conducted to propose techniques aimed at addressing this issue. Examples include techniques based on using filters [1,2,3,4,5,6,7], frequency domain signal processing [8], an independent component analysis [9,10], a Fourier series analysis [11], a time–frequency spectral analysis [12], and modulating techniques [13,14].

The technique of modulation was proposed by Sinchai, S. et al. [13]. Based on the study in this article, the frequency components of the moving artifact signal and the measured PPG signal are in the same frequency band. Therefore, in this work, amplitude modulation (AM) is applied. By using AM, both red and IR components of the measured PPG signals are modulated using two different carrier signals. As a result, the frequency spectra of the moving artifact, red PPG, and IR PPG signals do not overlap. The clean version of the red and IR PPG signals can be retrieved, leading to a higher accuracy in SpO_2_ measurement. To validate the commercial feasibility of this principle [13], the concept presented in this work is practically implemented using a microcontroller as proposed by Kainan, P. et al. [14]. In the conventional SpO_2_ measurement technique, the conventional method uses two out-of-phase driving pulse signals to alternatively drive red and IR light sources [15,16]. As a result, the red and IR PPG signals are time division multiplexing (TDM) signals. Compared to the technique of [13], the obtained non-overlap frequency bands of the moving artifact, the red PPG, and the IR PPG signals are considered frequency division multiplexing (FDM) signals.

In the last decade, several studies on pulse oximeters based on microcontrollers have been proposed [17,18,19,20], including a handheld pulse oximeter based on Raspberry Pi B+ [17], a pulse oximeter device utilizing a PIC microcontroller [18], a pulse oximeter incorporating a recycled SpO_2_ sensor and Arduino microcontroller [19], and a blood oxygen monitor based on Atmel ATmega [20]. Despite using different microcontrollers, these works [17,18,19,20] relied on conventional techniques for SpO_2_ measurements.

In this research, a similar principle to [6] was applied to develop a technique for SpO_2_ measurement. In the field of telecommunications, transmitting multiple modulated signals is known as frequency division multiplexing, and quadrature multiplexing is a specific type of multiplexing for two AM signals. Therefore, in this research, the quadrature multiplexing technique is used to combine the red PPG and IR PPG signals. The proposed technique is implemented using a microcontroller to validate the theoretical concept. In the microcontroller implementation, all signals are digitized. To obtain two AM signals, where the information signals are the red PPG and the IR PPG signals, sinusoidal carrier signals are required. Typically, a lookup table is used to generate a sinusoid sequence by storing the desired values for the sinusoid waveform in the memory. However, this method requires a large memory usage for high-precision requirements. In this work, an alternative technique is utilized. A digital sinusoid waveform is generated using a second-order difference equation, where only two initial conditions need to be stored in the memory. In this study, the proposed technique aims to be applied to a wearable device. Currently, most commercial wearable devices use the conventional technique for SpO_2_ measurement. Therefore, the focus of this study was to compare the proposed technique with the conventional method. Additionally, considering a wearable scenario, the device is used both indoors and outdoors. Different levels of light intensity were set to simulate indoor and outdoor conditions and evaluate the performance of both the proposed technique and the conventional method.

The organization of this research is as follows: Section 1 provides an introduction, and Section 2 briefly reviews the basic principle of SpO_2_ determination, followed by discussions on amplitude modulation and quadrature multiplexing in Section 3. Section 3 also presents the proposed pulse oximeter based on quadrature multiplexing and the signal processing procedure for microcontroller implementation. The experimental results of the proposed technique, along with a comparison of accuracy with a commercial pulse oximeter, are illustrated in Section 4. Finally, the limitations and the conclusions are drawn in Section 5 and Section 6, respectively.

## 2. Blood Oxygen Saturation

### 2.1. Pulse Oximetry

Pulse oximetry was invented in 1974 by bioengineer Takuo Aoyagi [21]. It is a non-invasive and useful technology for monitoring the functional oxygen saturation of hemoglobin in arterial blood (SaO_2_). During the COVID-19 pandemic, this device became an important tool for self-monitoring oxyhemoglobin saturation to detect hypoxia in infected patients at an early stage.

The arterial hemoglobin oxygen saturation (SaO_2_) is related to variation in oxyhemoglobin (HbO_2_), ∆HbO_2_, and variation in deoxyhemoglobin (Hb), ∆Hb, as given by [22,23]:SaO_2_ = ∆HbO_2_/(∆HbO_2_ + ∆Hb).(1)

It has been observed that oxyhemoglobin and deoxyhemoglobin absorb red and IR light differently. Specifically, HbO_2_ absorbs more IR light than red light, while Hb absorbs more red light than IR light. Based on this difference in light absorption properties, a pulse oximeter emits two lights: red light at a wavelength of 660 nm and IR light at a wavelength of 940 nm, using a pair of light-emitting diodes. These lights are transmitted through the finger and detected by a photodiode. The detected signal is known as a photoplethysmographic (PPG) signal. Figure 1 illustrates the acquisition of a PPG signal using a commercial pulse oximeter [6]. As the light penetrates through the finger, it is influenced by the pulsating and non-pulsating arterial blood volume caused by the cardiac cycle and other relatively constant parameters such as venous blood, skin, fat, bone, etc. As a result, these parameters contribute to the AC component and DC component in the PPG signal. The general waveform of a PPG signal is demonstrated in Figure 2 [24].

The pulse oximeter utilizes the amplitude of absorbances (A) to calculate the modulation ratio R. R is defined as the double ratio of the pulsatile and non-pulsatile components of red-light absorption to IR-light absorption, where its mathematical expression is [A_RED(AC)_/A_RED(DC)_]/[A_IR(AC)_/A_IR(DC)_]. In cases of low arterial oxygen saturations where Hb is high, the relative change in amplitude of the red-light absorbance is greater than that of the IR-light absorbance, resulting in a higher R value. Conversely, at higher oxygen saturations, the relative change in amplitude of the IR-light absorbance is greater than that of the red-light absorbance, leading to a lower R value.

For simplicity, the absorbance value, A, is calculated as the ratio of the AC component to the DC component of the PPG signal in the form of electrical voltage. The ratio R is then determined by
R = [V_RED(AC)_/V_RED(DC)_]/[V_IR(AC)_/V_IR(DC)_].(2)

Due to the dissimilarity of each human body and the different manufacturing materials used in each pulse oximeter brand, it is necessary to normalize this ratio. Normalizing the ratio allows for the consideration of factors related to human and instrumental diversities on the same scale. A pulse oximeter can only determine arterial SaO_2_ by measuring changes in absorbance over time. Therefore, a sufficient pulse of a PPG signal is required to accurately estimate the true SaO_2_. The pulse oximeter reading of oxygen saturation is referred to as SpO_2_, and under the same normalization of the ratio, the calculation of SpO_2_ is defined by the following empirical calibration equation [25]:SpO_2_ = 110 − 25R(3)
where the ratio R is determined by (2). It should be noted that a normal oxygen saturation level is between 87% and 97%, and any level below 70% should not be considered for making clinical decisions [26]. The provided expression ensures accuracy in measuring SpO_2_ within 2.5%. Additionally, when using a pulse oximeter to measure SpO_2_, the measurement region should have sufficient blood perfusion; otherwise, the result may not be detected accurately or may yield an inaccurate reading.

### 2.2. Amplitude Modulation (AM) and Quadrature Multiplexing

Modulation is a technique used to shift the frequency components of the information signal to higher frequencies. In telecommunication systems, its primary purpose is to generate a signal that can be transmitted through the channel. However, in the application of pulse oximeters, modulation is required to avoid any interference caused by the motion artifact signal [13].

For analog modulation, amplitude modulation is a simple and suitable technique for the application of pulse oximeters. The signals involved in the modulation process are the carrier signal, which is a high-frequency sinusoidal signal A_c_cos(ω_c_t), and the information signal, which in this case is the PPG signal. Due to the nature of the PPG signal, it contains both a DC component and an AC component (as shown in Figure 2). When it is modulated with the carrier signal, the standard AM signal is obtained, which can be expressed as [27]:ϕ_AM_(t) = A_c_(k + m(t))cos(ω_c_t)(4)
where k and m(t) are the DC and AC components of the PPG signal, respectively. As can be seen from (4), the amplitude of the carrier signal, A_c_(k + m(t)), is varied by the instantaneous value of the information signal.

To determine the SpO_2_ value of the pulse oximeter, two PPG signals are required. One signal is the red PPG signal related to deoxyhemoglobin (Hb), and the other is the infrared PPG signal related to hemoglobin (HbO_2_). Therefore, two AM signals related to the two PPG signals are
ϕ_AM_RED__(t) = A_c_RED__(k_RED_ + m_RED_(t))cos(ω_c_RED__t)(5)
and
ϕ_AM_IR__(t) = A_c_IR__(k_IR_ + m_IR_(t))cos(ω_c_IR__t)(6)
where ω_c_RED__ and ω_c_IR__ are the frequencies of the carrier signals employed for modulating the red PPG signal and the IR PPG signal, respectively. Let ϕ_AM_PPG__(t) be the combination of these AM signals, which is given by
ϕ_AM_PPG__(t) = ϕ_AM_RED__(t) + ϕ_AM_IR__(t)                         = A_c_RED__(k_RED_ + m_RED_(t))cos(ω_c_RED__t) + A_c_IR__(k_IR_ + m_IR_(t))cos(ω_c_IR__t).(7)

In the frequency domain, the Fourier transform of ϕ_AM_PPG__(t), Φ_AM_PPG__(ω), can be expressed as
Φ_AM_PPG__(ω) = Φ_AM_RED__(ω) + Φ_AM_IR__(ω)               = A_c_RED__k_RED_π[δ(ω − ω_c_RED__) + δ(ω + ω_c_RED__)]                  + (A_c_RED__/2)[M_RED_(ω − ω_c_RED__) + M_RED_(ω + ω_c_RED__)]           + A_c_IR__k_IR_π[δ(ω − ω_c_IR__) + δ(ω + ω_c_IR__)]              + (A_c_IR__/2)[M_IR_(ω − ω_c_IR__) + M_IR_(ω + ω_c_IR__)].(8)

By selecting the proper carrier frequencies ω_c_RED__ and ω_c_IR__to ensure that the frequency spectra of Φ_AM_RED__(ω) and Φ_AM_IR__(ω) do not overlap, the demodulation process to recover the PPG signals can be achieved using two bandpass filters and two envelope detectors [13].

From a telecommunication perspective, the signal given by (8) is considered as an FDM (frequency division multiplexing) signal. Quadrature multiplexing is another technique based on amplitude modulation used for signal multiplexing, allowing two information signals to be combined. Therefore, quadrature multiplexing can be applied to multiplex the red and infrared PPG signals of a pulse oximeter. The major advantage of quadrature multiplexing is that the carrier signals have the same frequency but are quadrature in phase; for example, when one carrier signal is cos(ω_c_t), then another carrier signal will be either cos(ω_c_t − π/2) or cos(ω_c_t + π/2). Quadrature multiplexing is applied to the red and infrared PPG signals; the block diagram of the multiplexing and demultiplexing process is demonstrated in Figure 3. In the multiplexing process, the carrier signals are cos(ω_c_t) and sin(ω_c_t), respectively, and (8) can be rewritten as
ϕ_AM_PPG__(t) = ϕ_AM_R__(t) + ϕ_AM_IR__(t)                    = A_c_R__(k_R_ + m_R_(t))cos(ω_c_t) + A_c_IR__(k_IR_ + m_IR_(t))sin(ω_c_t)(9)
where the Fourier transform of this signal is
Φ_AM_PPG__(ω) = Φ_AM_R__(ω) + Φ_AM_IR__(ω)                           = A_c_R__k_R_π[δ(ω − ω_c_) + δ(ω + ω_c_)] + (A_c_R__/2)[M_R_(ω − ω_c_) + M_R_(ω + ω_c_)]                             − jA_c_IR__k_IR_π[δ(ω − ω_c_) − δ(ω + ω_c_)] − j(A_c_IR__/2)[M_IR_(ω − ω_c_) − M_IR_(ω + ω_c_)].(10)

From (10), since both AM signals employ the same carrier frequency, the frequency spectra of the Φ_AM_R__(ω) and Φ_AM_IR__(ω) completely overlap. In the demodulator, bandpass filters and envelope detectors cannot be applied to retrieve the red and infrared PPG signals. Instead, the demultiplexing process requires synchronous demodulation, which consists of a multiplier and a lowpass filter. Synchronous demodulation allows for the regeneration of carrier signals at the demodulator with an identical frequency and phase to those of the modulator. Based on this assumption, ϕ_D_R__(t) and ϕ_D_IR__(t) are given by (11) and (12), respectively.
ϕ_D_R__(t) = ϕ_AM_PPG__(t)cos(ω_c_t)                        = [A_c_R__(k_R_ + m_R_(t))cos(ω_c_t) + A_c_IR__(k_IR_ + m_IR_(t))sin(ω_c_t)]cos(ω_c_t)                     = [A_c_R__(PPG_R_(t))cos(ω_c_t) + A_c_IR__(PPG_IR_(t))sin(ω_c_t)]cos(ω_c_t)                           = (A_c_R__/2)PPG_R_(t) + (A_c_R__/2)PPG_R_(t)cos(2ω_c_t) + (A_c_IR__/2)PPG_IR_(t)sin(2ω_c_t)(11)
ϕ_D_IR__(t) = ϕ_AM_PPG__(t)sin(ω_c_t)                        = [A_c_R__(k_R_ + m_R_(t))cos(ω_c_t) + A_c_IR__(k_IR_ + m_IR_(t))sin(ω_c_t)]sin(ω_c_t)                     = [A_c_R__(PPG_R_(t))cos(ω_c_t) + A_c_IR__(PPG_IR_(t))sin(ω_c_t)]sin(ω_c_t)                            = (A_c_R__/2)PPG_R_(t)sin(2ω_c_t) + (A_c_IR__/2)PPG_IR_(t) + (A_c_IR__/2)PPG_IR_(t)cos(2ω_c_t)(12)

When ϕ_D_R__(t) and ϕ_D_IR__(t) signals are passed through the lowpass filter, the demultiplexed signals are obtained as follows:ϕ_PPG_R__(t) = (A_c_R__/2)PPG_R_(t)        = (A_c_R__/2)(k_R_ + m_R_(t))(13)
and
ϕ_PPG_IR__(t) = (A_c_IR__/2)PPG_IR_(t)         = (A_c_IR__/2)(k_IR_ + m_IR_(t)).(14)

Based on the mathematical analysis shown above, it is evident that quadrature multiplexing can be utilized to combine the red and infrared PPG signals. Despite the frequency spectra of the two AM signals falling within the same bandwidth, the phase–quadrature property enables the demultiplexing and subsequent demodulation of the red and infrared PPG signals. From the demodulated signals, the AC components of the red and infrared PPG signals are extracted to determine the SpO_2_ level.

Recently, the concept of applying AM for calculating SpO_2_, as proposed by Sinchai, S. et al., has been implemented in a commercial device [14]. In their research [14], the pulse oximeter was implemented using a microcontroller, and the carrier signal was digitally generated. This implementation serves as a basis for applying the proposed technique in this research. The discussion of this implementation will be provided in the next section.

## 3. The Proposed System Based on Quadrature Multiplexing

Based on the quadrature multiplexing technique, the proposed system for the pulse oximeter is demonstrated in Figure 4. The implementation of this pulse oximeter is based on the microcontroller, where the technique of [14] is adapted in this article. The realized system, utilizing the microcontroller (ESP32), processes the signal digitally. The signal processing procedure of the proposed pulse oximeter is illustrated by the flowchart in Figure 5.

In Figure 5, two discrete sequences, namely cosine and sine sequences, are generated utilizing the zero-input response of the second-order difference equation as given by [28]
y(n) − 2cos(θ)y(n − 1) + y(n − 2) = 0(15)

For the cosine sequence y(n) = Acos(θn), the initial conditions are y(−1) = Acos(-θ) and y(−2) = Acos(−2θ). To obtain the sine sequence y(n) = Asin(θn), the initial conditions are y(−1) = Asin(−θ) and y(−2) = Asin(−2θ). The parameter A is the peak amplitude and θ is a digital frequency, 0 < θ < π. The discrete sequences can be converted to the analog signals using the sampling rate of f_s_ Hz where the analog frequency ω is θ/f_s_ rad/s.

The discrete sinusoidal sequences are transmitted to the microcontroller’s D/A ports to generate the analog sinusoidal carrier signals. These analog carrier signals are used to drive the red and IR LEDs of the pulse oximeter probe. The resulting FDM signal, which is the multiplexing of the two AM signals, is then sent to the A/D port and converted into a discrete sequence. The discrete FDM sequence is multiplied with the discrete cosine and sine sequences separately. After applying lowpass filtering to the obtained results, the light absorbance ratios of the red and IR components are determined. Subsequently, the SpO_2_ level is calculated. To assess the accuracy of the proposed pulse oximeter, the relative accuracy is employed, as shown in (16).
Relative accuracy = (measured SpO_2_ level/reference SpO_2_ level) × 100%(16)

The relative accuracy, derived from (16), is obtained by taking the ratio of the measured SpO_2_ level to the reference SpO_2_ level obtained from the conventional pulse oximeter and multiplying it by 100%.

## 4. Experimental Results

To validate the proposed pulse oximeter, an experimental implementation of the technique using the ESP32 microcontroller is conducted. The analog frequency of both carrier signals is set to 90 Hz. The ESP32 generates the cosine and sine sequences based on the second-order difference equation provided in (15). The parameters are configured as follows: amplitude A: 3.3V, digital frequency θ: 0.5522 radians, and sampling frequency f_s_: 1024 Hz. The two generated digital sinusoidal signals are stored as numeric values in the memory and can be observed in Figure 6. Both carrier signals are the same frequency but differ in phase by |π/2|. The analog versions of the cosine and sine signals drive the red-light source and the infrared-light source of the pulse oximeter probe, respectively, and the resulting FDM signal is obtained as the output of the amplifier.

Firstly, the simulated PPG signals are generated using the Fluke SPOTLight SpO_2_ Functional Tester. An example of the FDM signal, which is a combination of two AM signals representing 100% SpO_2_ levels for the red and infrared PPG signals, is depicted in Figure 7 (top) and Figure 7 (bottom) in the time domain and frequency domain, respectively. In Figure 7 (bottom), it can be observed that the two PPG signals are frequency-shifted to be centered at the same frequency, demonstrating the characteristic of quadrature multiplexing. Since the controlling signals are generated by the microcontroller and transmitted through the D/A port, the obtained carrier signal frequency may slightly deviate from the set value of 90 Hz.

In the demodulation process, a synchronous demodulator is required, where the cosine and sine sequences generated by the ESP32 are utilized to demodulate the red PPG signal and the IR PPG signal. Upon acquiring the quadrature AM signal from the amplifier circuit, it is multiplied with the cosine and sine sequences. The multiplication outputs are then subjected to lowpass filtering to obtain the two components of the PPG signals: red and IR. Figure 8 presents examples of demodulated PPG signals, generated by the Fluke SPOTLight SpO_2_ Functional Tester, corresponding to SpO_2_ levels of 70%, 85%, and 100%.

Subsequently, the Fluke SPOTLight SpO_2_ Functional Tester is used to simulate PPG signals at various SpO_2_ levels, specifically at 70%, 75%, 80%, 85%, 90%, 95%, 97%, 98%, 99%, and 100%. To estimate different SpO_2_ levels more accurately, the original empirical calibration illustrated in (3) is slightly tuned. This is because the device used to generate the relationship in (3) differs from the one implemented in this study. Each level of the PPG signals is tested five times to measure the SpO_2_ level using the proposed technique. Based on these data, the linear regression analysis is conducted to determine the relationship between the measured SpO_2_ level and the proposed technique, as expressed by (17).
SpO_2_ level = 112.5833 − 31.0213R(17)

Using Equation (17) to determine the SpO_2_ level of these PPG signals, the plot of the percentage of relative error is shown in Figure 9, with an average value of approximately 1.1%.

In the accuracy comparison, only the conventional method is compared to the proposed technique because, as mentioned earlier, most mercantile contemporary wearable pulse oximeters adopt the traditional approach. This is because the conventional method is easy to implement. Furthermore, various SpO_2_ IC sensors adopting the conventional method have been commercially marketed. Even the classic one is simple, but it faces the problem of light variate interference where the proposed technique is better immunized to this problem. Nonetheless, before performing the accuracy comparison with the conventional method, the PPG signal quality of the proposed technique is verified by utilizing the SpO_2_ level as a criterion. Since the proposed technique is developed from the method of [14], the SpO_2_ values yielded by [14] are thus used as a reference in the PPG signal quality evaluation. This evaluation is performed using the Wilcoxon–Mann–Whitney test as the SpO_2_ results obtained from the method of [14] are independent of the SpO_2_ results produced by the presented approach [29]. The SpO_2_ results of the proposed method are calculated from the red and IR PPG signals of 20 volunteers. These red and IR PPG signals are collected in a similar condition conducted in [4]; namely, a room is adjusted to have ambient light that barely changes. Moreover, the volunteers are asked to stay still while the measurement is taking place for 5 seconds each time for a total of five times for each volunteer. The SpO_2_ results of the proposed approach and the method of [4] are displayed in Table 1.

To evaluate the PPG signal quality of both techniques by the Wilcoxon–Mann–Whitney test, two hypotheses are stated as follows where H_0_ is the null hypothesis and H_1_ is an alternative hypothesis.

**H_0_:** 
*The PPG signals from the proposed technique and the method of [14] are different.*


**H_1_:** 
*The PPG signals from the proposed technique and the method of [14] are not different.*


By inserting the SpO_2_ results from Table 1 into statistical software to calculate the Wilcoxon–Mann–Whitney two-tailed test at the 5% significance level (α = 0.05), the test result yields a rejection of the null hypothesis, H_0_. With this test result, it is conclusive that the PPG signal quality of the proposed technique does not differ from the PPG signal quality of the method proposed in [14]. This PPG signal quality test is conducted to confirm that the PPG signals acquired from the offered approach are valid for use in further studies. Nevertheless, the proposed technique is distinct from the approach of [14] in terms of bandwidth usage. The proposed technique adopts a lesser bandwidth than that of using the method of [14].

Later, abrupt light change affecting the PPG signals is studied in this work. The PPG signals are measured under three different light intensities of the environment: 200 (indoor), 950 (outdoor on an overcast day), and 2200 (outdoor on a sunny day) Lux, respectively. For each light intensity level, three conditions are of interest during the PPG signal measurement: hand still, shadow movement over the hand, and hand shaking. These scenarios are experimented on a 1 m x 1 m wood table and the artificial light source is placed directly facing down on the pulse oximeter, as depicted in Figure 10. It is important to note that for shadow movement over the hand, an object, such as a piece of paper, is used to create waving shadows in both vertical and horizontal directions over the hand during testing. The first scenario is designed to demonstrate that under different light intensities, even a high intensity but having no swift change, the PPG signals are less affected when utilizing the proposed technique. On the contrary, high light intensities distinctly affect the PPG signals when using the conventional method. The other two scenarios are designed to show that sudden light change caused by any movements significantly deteriorates the PPG signals when using the conventional method but not for the proposed technique. The movement in the second condition imitates a situation in which people walk around and cause sudden light changes, which affect the amount of light absorption indirectly through variant shadow intensities. For the movement produced in the third condition, the hand is intentionally moved, and abrupt light variations directly affect the amount of light absorption.

Based on these situations, the PPG signals are measured from the same previous 20 volunteers using both the proposed technique and the conventional technique. Each volunteer wears the devices for the proposed technique and the conventional technique on the index and middle fingers of the same hand. Examples of the detected red and IR PPG signals for each condition, based on the proposed technique and the conventional technique, are shown in Figure 11 and Figure 12, respectively.

The result of the experiment is shown in Table 2. Overall, the proposed technique outperforms the conventional technique. It demonstrates consistency in determining the SpO_2_ level across most light intensity levels and testing conditions. Moreover, the relative error obtained from the proposed technique is lower than that of the conventional method. In the case of the conventional technique, it is evident that a higher light intensity, light variation, and hand movement negatively impact its performance. Some PPG signals measured using the conventional technique fail to determine the SpO_2_ levels. Based on this study, the proposed technique shows promise for application in wearable devices with SpO_2_ measurement capabilities, even under conditions of a high light intensity and hand movement.

Due to an inability to imitate the same environments and similar algorithms to those used in [1,2,3,4,5,6,7,8,9,10,11,12,13,14], it was considered that reproducing their experiments may not produce good results. Instead of comparing the accuracy, the time complexity is applied in this work to gauge the amount of the computer time to recover the PPG signals before performing the SpO_2_ calculation. Generally, the Big O notation, O(·), is an instrument for explaining the time complexity of algorithms [30]. Based on the literature studies of [1,2,3,4,5,6,7,8,9,10,11,12,13,14], their time complexity is grouped into three kinds, O(N) [1,2,3,4,5,13,14], O(NlogN) [8,9,11,12], and O(N^2^) [6,7,10], where N is the sample size. For this research, its time complexity is classified into the group of O(N). The three kinds of time complexity are graphically illustrated in Figure 13, where the solid line represents the O(N) of [1,2,3,4,5,13,14] and the dash-dotted line refers to the O(N) of the proposed technique. Moreover, the dotted line indicates the O(NlogN) of [8,9,11,12] and the dashed line shows the O(N^2^) of [6,7,10]. As can be observed in Figure 13, the running time complexity of O(N) linearly increases when the number of N increases. Conversely, both O(NlogN) and O(N^2^) nonlinearly rise in their running time complexity when their sample sizes move up. Obviously, this work is superior to [6,7,8,9,10,11,12] in terms of computational complexity. After delving into the research details of [1,2,3,4,5], they have a few extra steps of updating their filter coefficients due to their adaptive manner. On the other hand, no additional step for the offered method is required because the filter coefficients are constant. This means that the running time complexity for [1,2,3,4,5] is greater than the proposed technique. Hence, it can be inferred that this research has an advantage over the research studies of [1,2,3,4,5], even being categorized in the same group of O(N). For the research works [13,14], their running time complexity is slightly higher than this work as they have an extra step of separating two PPG signals from each other.

## 5. Limitations

Since this research is still in the stage of laboratory testing, the wearable device, all the circuits, and the ESP32 microcontroller are not mounted in the same package. Moreover, the wearable device is just a simple probe, and it does not have a wireless interface. Hence, the probe needs to connect to all the circuits including the ESP32 controller in a protoboard. With this purpose, the conducted experimental scenarios are thus limited to only sitting on the chair close to the testing terminal. Some experimental scenarios like walking or running are possible to conduct but difficult to perform. This is because the human arm on which the wearable is worn is swung by nature when walking or running, resulting in all the wiring connections in the protoboard being disconnected. With this restriction, the PPG signal cannot be measured continuously for long enough. The acquired PPG signal may not be good enough to provide significant components to compute the SpO_2_ level. Nonetheless, this work has attempted to conduct an experimental scenario introducing naturalistic motion artifacts caused by light change due to walking in place under this work’s constraint performed by two subjects. Each one walks in place under a light intensity of 200 Lux with the same time duration proposed in Section 4 (repeated five times). The obtained red and IR PPG signals of both are then converted into the SpO_2_ levels. The resulting average SpO_2_ level yields a mean relative error roughly 1.6% greater than the worst case of the proposed technique (3.1%) illustrated in Table 2. Because the experimental scenario, walking in place, is tested with a small number of volunteers, the obtained low relative error cannot yet conclude that the proposed method is marginally affected by the realistic motion artifacts. To confirm the performance of the proposed technique, a greater number of volunteers are required to experiment after the wearable device, all the circuits, and the ESP32 microcontroller are mounted in the same package.

## 6. Conclusions

This research proposes a pulse oximeter utilizing the ESP32 microcontroller, employing the technique of quadrature multiplexing-based amplitude modulation. The red and IR light sources in the pulse oximeter probe, connected in a common anode configuration, are controlled by sinusoidal signals of the same frequency (90 Hz) but with a quadrature phase. The detected signal from the light sensor is in the form of FDM, combining two PPG AM signals. In the demodulation process, two synchronous detectors directly recover the red PPG and infrared PPG signals. Compared to the conventional technique, the proposed technique in this research demonstrates an improved performance. The overall relative error obtained from the experiment is approximately 3.1% or less. The light intensity, light variation, and hand movement have minimal impact on the performance of the proposed technique. Moreover, the time complexity of the proposed technique is lower than that of other existing methods. Therefore, the proposed technique is well suited for application in wearable devices with SpO_2_ measurement capabilities, even under conditions of a high light intensity and hand movement.

## Figures and Tables

**Figure 1 sensors-23-06106-f001:**
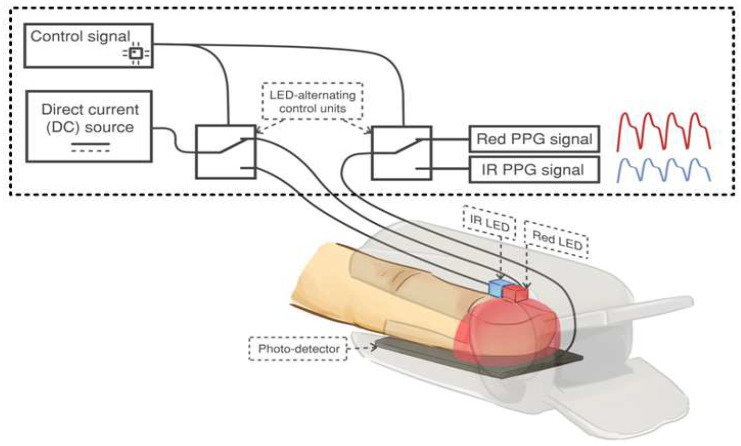
The acquisition of a PPG signal using a commercial pulse oximeter. Adapted with permission from Ref [13]. Copyright 2018, the Institute of Electrical and Electronics Engineers, IEEE.

**Figure 2 sensors-23-06106-f002:**
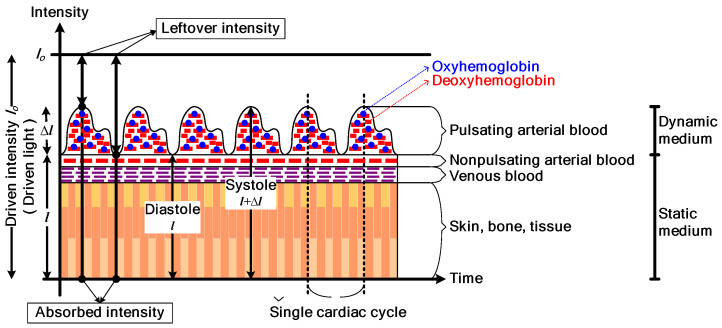
The general waveform of a PPG signal. Reprinted with permission from Ref [24]. Copyright 2018, the Institute of Electrical and Electronics Engineers, IEEE.

**Figure 3 sensors-23-06106-f003:**
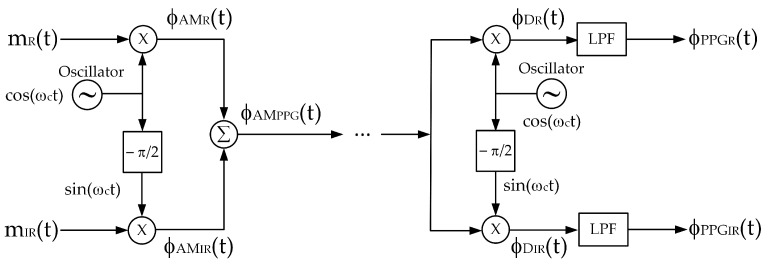
A block diagram of quadrature multiplexing.

**Figure 4 sensors-23-06106-f004:**
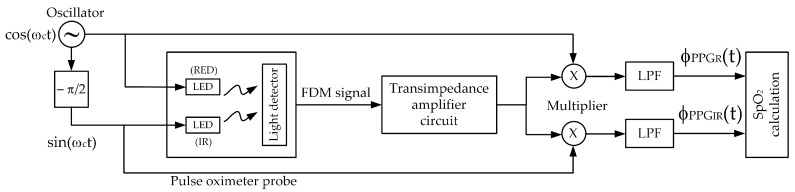
The block diagram of the proposed system.

**Figure 5 sensors-23-06106-f005:**
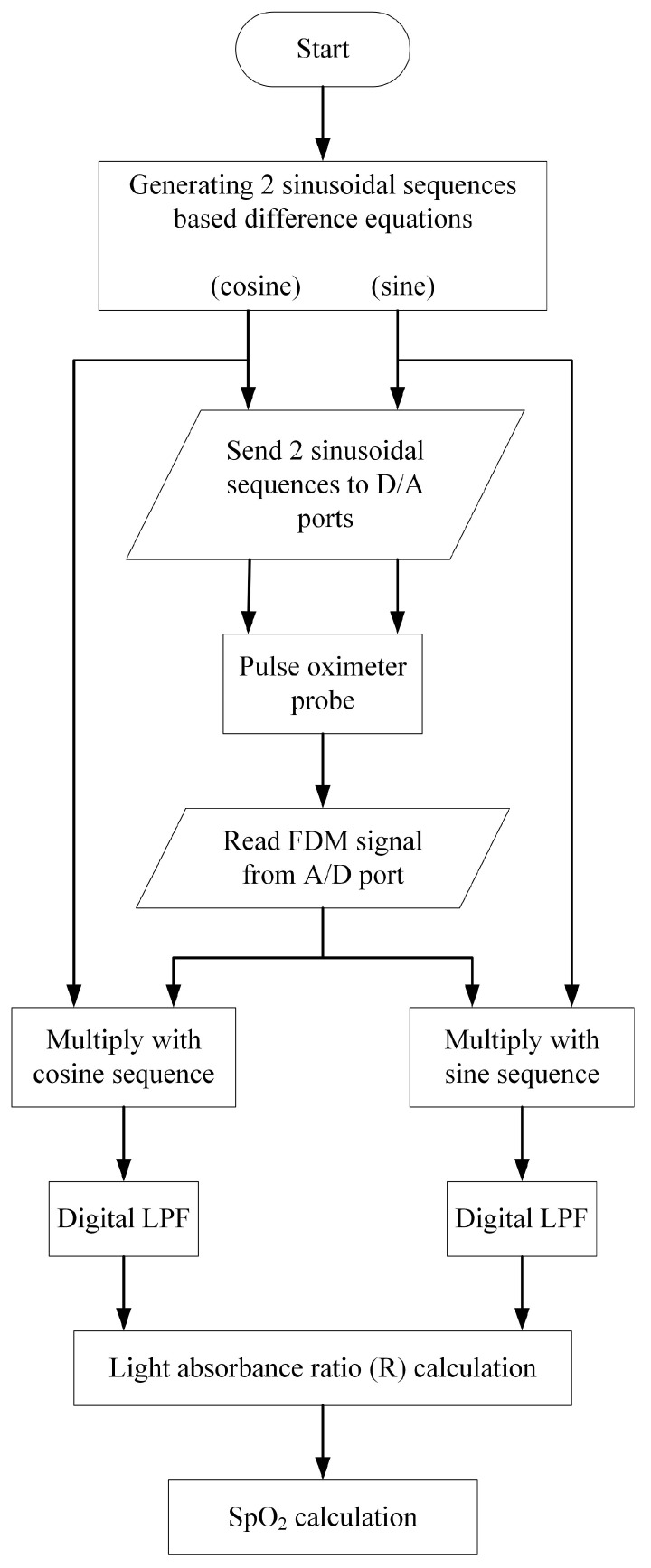
Flowchart of signal processing of the proposed system based on the microcontroller (ESP32).

**Figure 6 sensors-23-06106-f006:**
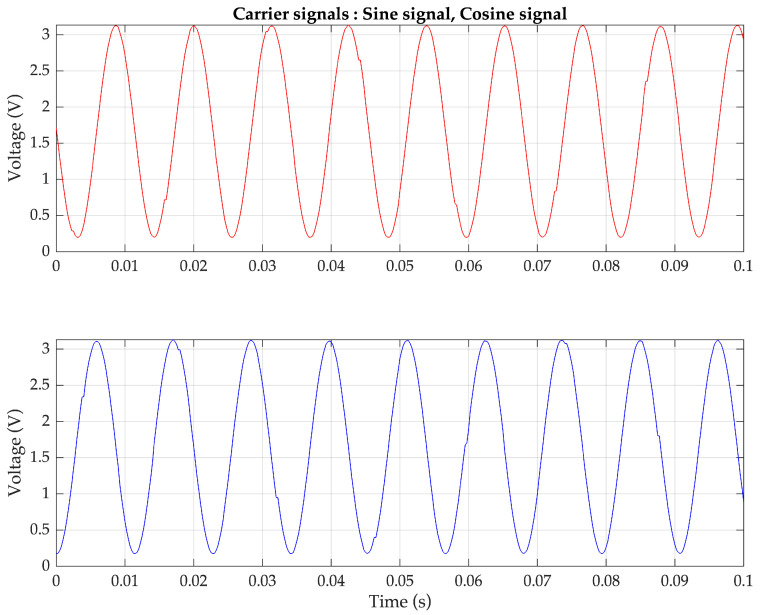
Sinusoidal sequences generated by the ESP32 using a difference equation (**top**) 90 Hz cosine sequence and (**bottom**) 90 Hz sine sequence for controlling two light sources.

**Figure 7 sensors-23-06106-f007:**
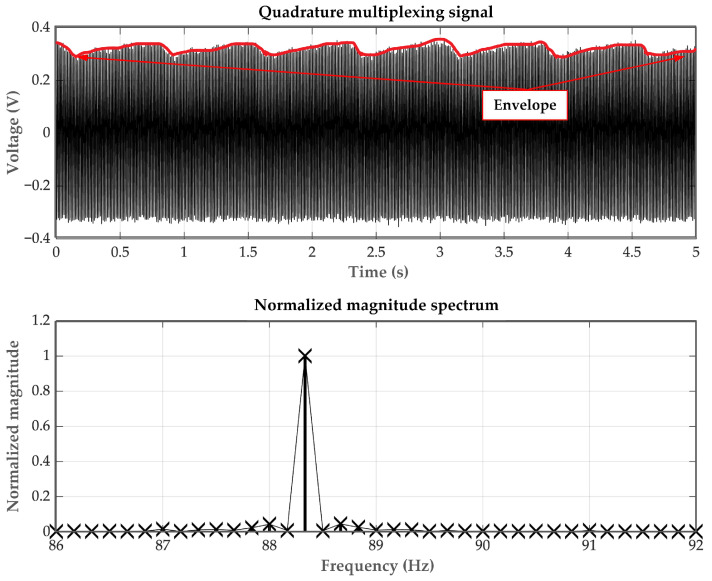
The AM-PPG signals (**top**) in the time domain and (**bottom**) in the frequency domain.

**Figure 8 sensors-23-06106-f008:**
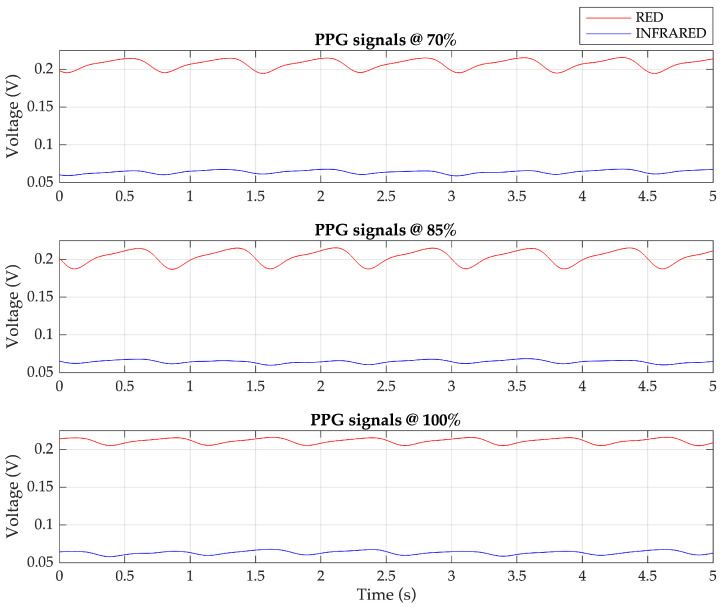
The demodulated PPG signals: red and infrared components for 70%, 85%, and 100% SpO_2_ levels.

**Figure 9 sensors-23-06106-f009:**
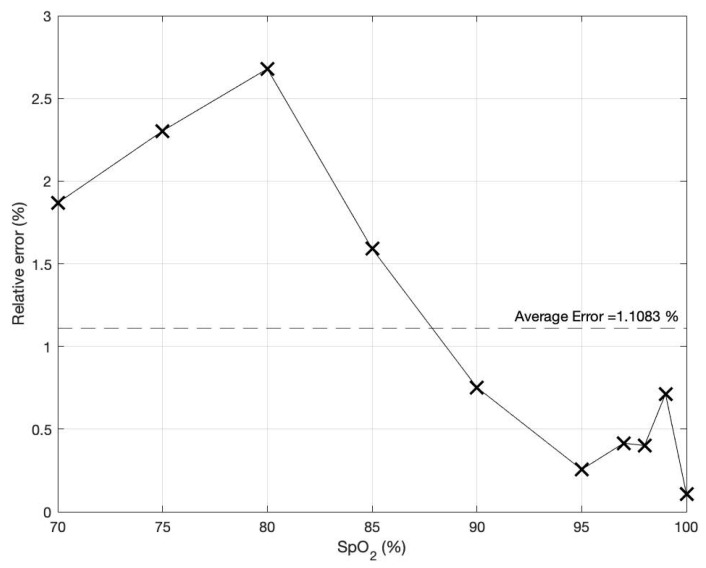
The percentage of relative error of the determined SpO_2_ level based on the proposed technique.

**Figure 10 sensors-23-06106-f010:**
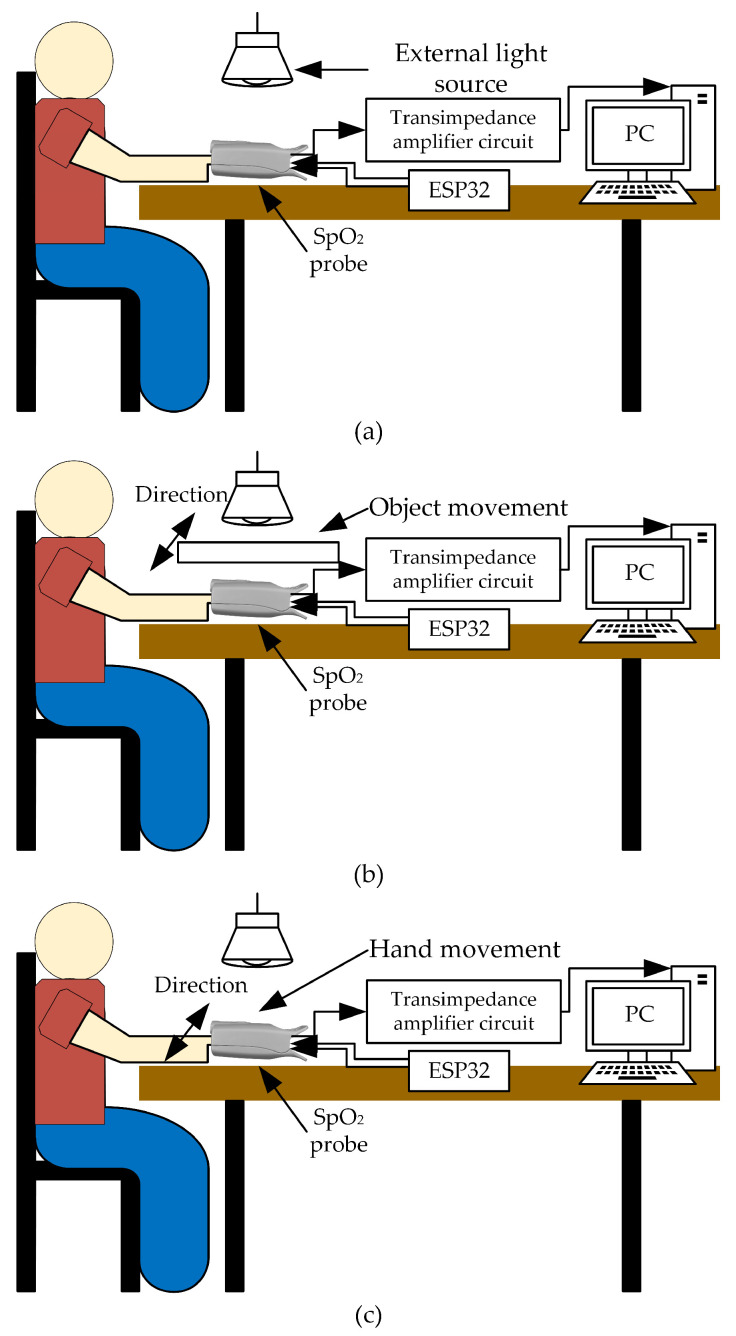
Testing scenarios: (**a**) hand still, (**b**) shadow movement over the hand, and (**c**) hand shaking.

**Figure 11 sensors-23-06106-f011:**
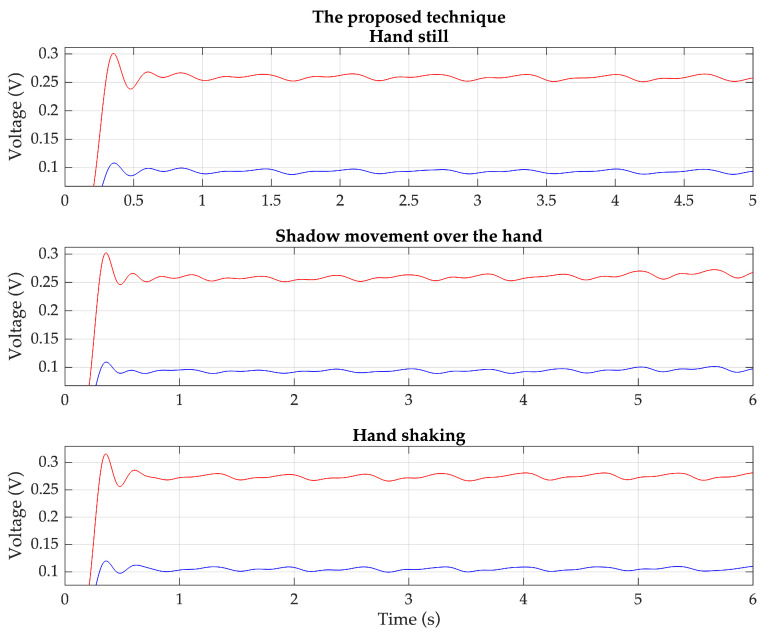
Example of the detected red (red line) and IR (blue line) PPG signals of each condition based on the proposed technique.

**Figure 12 sensors-23-06106-f012:**
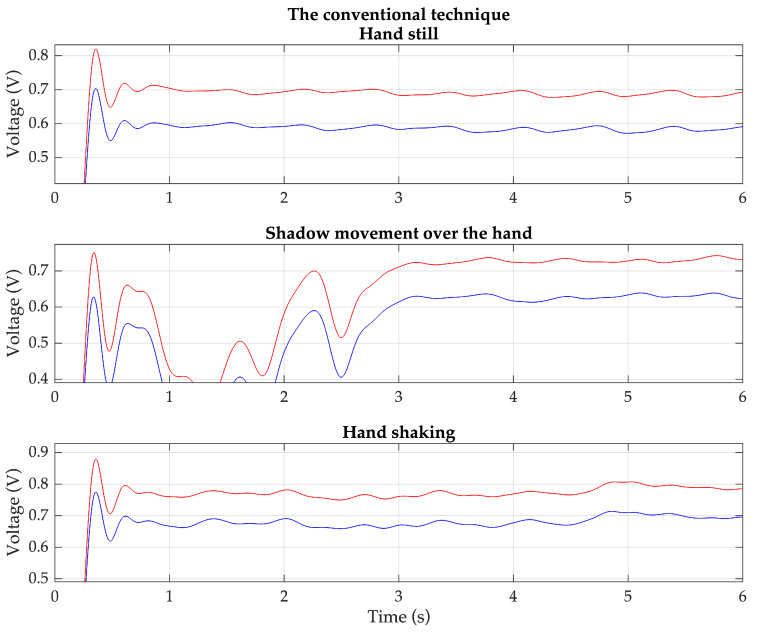
Example of the detected red (red line) and IR (blue line) PPG signals of each condition based on the conventional technique.

**Figure 13 sensors-23-06106-f013:**
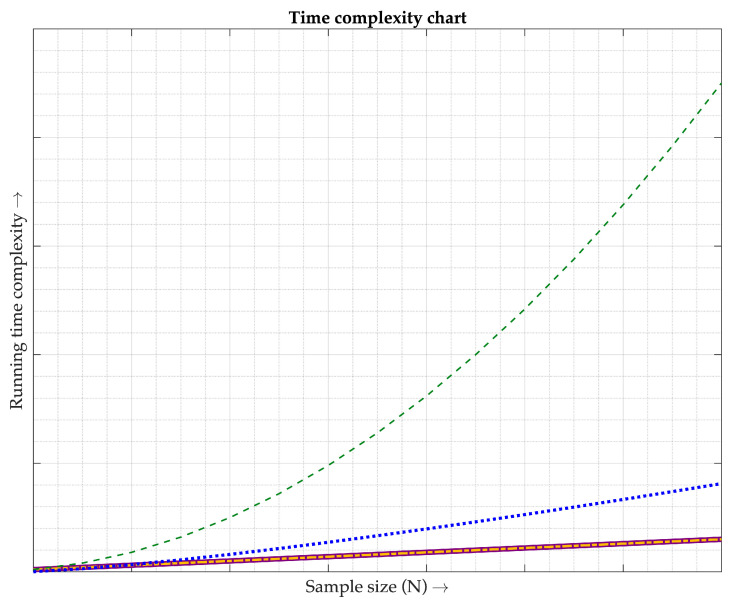
Comparison of time complexity. Each line type is described as follows. The purple solid line shows the time complexity of the methods used in [1,2,3,4,5,13,14], classified as O(N). The gold dash-dotted line illustrates the time complexity of the proposed technique, classified as O(N). The blue dotted line depicts the time complexity of the methods utilized in [8,9,11,12], classified as O(NlogN). The green dashed line demonstrates the time complexity of the methods employed in [6,7,10], classified as O(N^2^).

**Table 1 sensors-23-06106-t001:** SpO_2_ results acquired from the proposed technique and the method of [14]. Reprinted with permission from Ref. [14]. Copyright 2022, the Institute of Electrical and Electronics Engineers, IEEE.

**The Proposed Technique**				**The Method of [14]**	
Test subject	SpO_2_ value	Test subject	SpO_2_ value	Test subject	SpO_2_ value
Volunteer 1	93.5930	Volunteer 11	95.6550	Subject 1	97.0484
Volunteer 2	95.8557	Volunteer 12	95.6676	Subject 2	97.7825
Volunteer 3	100	Volunteer 13	97.6127	Subject 3	98.7652
Volunteer 4	94.5019	Volunteer 14	94.5412	Subject 4	99.0230
Volunteer 5	96.8928	Volunteer 15	96.0274	Subject 5	98.0114
Volunteer 6	99.0326	Volunteer 16	95.0141		
Volunteer 7	99.9118	Volunteer 17	96.2826		
Volunteer 8	92.1887	Volunteer 18	95.0588		
Volunteer 9	95.9125	Volunteer 19	97.2181		
Volunteer 10	97.5498	Volunteer 20	95.6176		

**Table 2 sensors-23-06106-t002:** Evaluation of accuracy of the proposed method.

Light Intensity (Lux)	Test Scenario	Method	Number of Valid Signals	Relative Error (%)
200	Hand still	Proposed	20	2.1755
		Conventional	20	5.3973
	Shadow movement over the hand	Proposed	20	2.0594
		Conventional	20	7.3954
	Hand shaking	Proposed	20	2.5517
		Conventional	19	7.5915
950	Hand still	Proposed	20	1.8475
		Conventional	19	14.9475
	Shadow movement over the hand	Proposed	20	1.8586
		Conventional	8	12.8583
	Hand shaking	Proposed	20	2.4604
		Conventional	11	15.6635
2200	Hand still	Proposed	18	2.3439
		Conventional	14	17.6307
	Shadow movement over the hand	Proposed	20	3.1034
		Conventional	7	15.5668
	Hand shaking	Proposed	20	2.9494
		Conventional	4	14.8089

## Data Availability

Not applicable.

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
