# Peer review of "Pulse Oximetry Based on Quadrature Multiplexing of the Amplitude Modulated Photoplethysmographic Signals"

_sensors, 2023, doi:10.3390/s23136106_

Round 1

Reviewer 1 Report

This project is good and needed, but some parts of it need to be re-examined by the author. For example: 1- The introduction of the article should be improved and better references should be used 2- All figures and diagrams must be presented with better quality. 3- The discussion of the article should be improved and more articles should be compared. 4- The article needs to be checked and improved in terms of grammar and English. 5- The referencing method should be observed and better and preferably more references should be used.  

The article needs to be checked and improved in terms of grammar and English

Author Response

Dear respected referee.

First of all, we would like to thank you for reviewing and providing helpful comments.

We have revised our paper in accordance with your comments. Please see the explanation below. For the revision in the paper, we use a red color font to indicate what has been corrected in the following comment no. 1 and 5. For comment no. 2, please consider the revised figures. Lastly, comment no. 3 and 4, the revisions is written by a purple colour font. A new Table has been added, marked as Table 1 and the previous Table 1 has been changed to Table 2, also in a purple colour font. Please see the attachment. Our responses are as follows.

Point1. The introduction of the article should be improved and better references should be used 

Response1: More explanation in the introduction has been added and the overall introduction has been paraphrased. In addition, we have rewritten the abstract.

Point2. All figures and diagrams must be presented with better quality. 

Response2: We have mended all the figures. All the figures have been reproduced in a better resolution.

Point3: The discussion of the article should be improved and more articles should be compared. 

Response3: 

We have compared our proposed technique with the method having the resembling architecture in terms of the signal quality. Besides, other existing methods were compared to our proposed technique by utilizing only the time complexity as a criterion.

The reasons are as follows.

- We could not reproduce the similar environment as well as setting parameters to those existing methods and we feel that if we did and their results would not be as good as shown in their reports, the reproduced experiments of our might not be fair for them.

- We could use the time complexity in comparison as we could analyse the algorithms reported in those other existing methods.

Further explanation and discussion have been provided in the revision paper.

Point4. The article needs to be checked and improved in terms of grammar and English. 

Response4: All sentences including phrases have been rechecked and paraphrased in the proper syntax structure.

Point5. The referencing method should be observed and better and preferably more references should be used. 

Response5: We have updated the referencing method abiding by the MDPI sensors template. Also, we have added more references relating works as well as theory.

Reviewer 2 Report

In this paper, the author proposed a pulse oximeter based on quadrature multiplexing of the AM-PPG signals. The paper is well wrote, however, the innovation is not stated clearly. And thus, I strongly suggest the authors add more related literatures to expound the present situation and trend of development, especially the quadrature multiplexing method in pulse oximetry. The following is the specific suggestions:

1  All equations should be cited their sources, except the formula created by the authors themselves.

2  It is not suitable that the level of SpO2 has two specific calculation formulas, those are Eq.3 and Eq.17. Or explain it.

3  What is the conventional technique? Is there any difference with your proposed technique? Please explain.

4  Figure 10 is not clear, and please give additional details about relationship of the testing scenarios and the sensing area, especially the light collecting face.

5  Figure 11 and 12 are not clear.

6  please compare the testing results of your strategy with other previous literatures.

7  please check the syntax structure.

Your paper is well written except for a few spelling mistakes.

Author Response

Dear respected referee.

First of all, we would like to thank you for reviewing and providing helpful comments.

My other authors and I have revised our paper in accordance with your helpful comments and attached the revision. For the revision in the paper, we use a blue colour font to indicate what has been corrected in the following comment no. 1,2,3, and 4. For comment no. 5, please consider the revised figures. Lastly, comment no. 6 and 7, the revisions are written by a purple colour font. A new Table has been added, marked as Table 1 and the previous Table 1 has been changed to Table 2, also in a purple colour font. Please see the attachment. Our responses are as follows.

Point1. All equations should be cited their sources, except the formula created by the authors themselves. 

Response1. We have added the references to the equations (1), (3), (4) and (15) and the remaining equations are created by the authors.

Point2. It is not suitable that the level of SpO2 has two specific calculation formulas, those are Eq.3 and Eq.17. Or explain it. 

Response2: Generally, a medical device is annually calibrated. The calibration is usually adjusted the coefficients of the equation (3) first rather than replacing a new hardware component because changing the new hardware component is more complicated than adjusting the coefficients. Besides, changing the new hardware component would have additional expenses.

Further explanation about the equations (3) and (17) is annexed in the revised paper.

Point3. What is the conventional technique? Is there any difference with your proposed technique? Please explain.

Response3: The conventional technique has been described and its difference from the proposed technique is mentioned.

Point4. Figure 10 is not clear, and please give additional details about relationship of the testing scenarios and the sensing area, especially the light collecting face. 

Response4: Figure 10 has been further explained and the additional details about the relationship of the testing scenarios and the sensing area have been fully explained in the revised paper.

Point5. Figure 11 and 12 are not clear.

Response5: The Figures 11 and 12 have been reproduced in a better resolution.

Point6. please compare the testing results of your strategy with other previous literatures. 

Response6: 

We have compared our proposed technique with the method having the resembling architecture in terms of the signal quality. Besides, other existing methods were compared to our proposed technique by utilizing only the time complexity as a criterion. The reasons are as follows.

- We could not reproduce the similar environment as well as setting parameters to those existing methods and we feel that if we did and their results would not be as good as shown in their reports, the reproduced experiments of our might not be fair for them.

- We could use the time complexity in comparison as we could analyse the algorithms reported in those other existing methods.

Further explanation and discussion have been provided in the revision paper.

Point7. please check the syntax structure. 

Response7: All sentences including phrases have been rechecked and paraphrased in the proper syntax structure. 

Reviewer 3 Report

Review Report

The authors have proposed the pulse oximeter using the ESP32 microcontroller where the technique of quadrature multiplexing based amplitude modulation is applied. The paper is well written and contains relevant and enough results to be considered. However, I would like to suggest the following very minor corrections before the acceptance of the manuscript for publication in the Sensors Journal.

1.     The authors should consider writing the entire in-text equations (in sections 2 and 3 e.g., line 103) in words not equation editor or any equation software.

2.     Please improve the resolution of Figures 11 and 12

Author Response

Dear respected referee.

First of all, we would like to thank you for reviewing and providing helpful comments.

My other author and I have revised our paper in accordance with your helpful comments and attached the revision. For the revision in the paper, we use a green colour font to indicate what has been corrected for comment no. 1. For comment no. 2, please consider the revised figures. Please see the attachment. Our responses are as follows.

Point1. The authors should consider writing the entire in-text equations (in sections 2 and 3 e.g., line 103) in words not equation editor or any equation software.

Response1: All equations have been written in-text in lieu of using either the equation editor or equation software.

Point2. Please improve the resolution of Figures 11 and 12

Response2:  The Figures 11 and 12 have been reproduced in a better resolution.

Note:-A new Table has been added, marked as Table 1 and the previous Table 1 has been changed to Table 2, also in a purple colour font.

Reviewer 4 Report

The article is devoted to a relevant topic - increasing the reliability of SpO2 level measurement. Despite of many research, there is no ideal solution exists. I would recommend to the authors in the introduction to review in more detail other methods for solving this problem (L44 - L50). Some references could be added during this process.

The article describes the developed technique in sufficient detail. The results and figures look good, except Figures 11 and 12, which are too compressed in a pdf file.

Why were precisely 200, 950, and 2200 Lux light intensities chosen?

Author Response

Dear respected referee.

First of all, we would like to thank you for reviewing and providing helpful comments.

My other author and I have revised our paper in accordance with your helpful comments and attached the revision. Please see the attachment. Our responses are as follows.

Point1. The article is devoted to a relevant topic - increasing the reliability of SpO2 level measurement. Despite of many research, there is no ideal solution exists. I would recommend to the authors in the introduction to review in more detail other methods for solving this problem (L44 - L50). Some references could be added during this process.

Response1: We have already added some more references and given more explanation (see L45-L95).

Point2. The article describes the developed technique in sufficient detail. The results and figures look good, except Figures 11 and 12, which are too compressed in a pdf file.

Response2: The Figures 11 and 12 have been reproduced in a better resolution. Please see the revised figures.

Point3. Why were precisely 200, 950, and 2200 Lux light intensities chosen?

Response3: The explanation for the question have been answered at L340-L358.

Note:-A new Table has been added, marked as Table 1 and the previous Table 1 has been changed to Table 2, also in a purple colour font.

Round 2

Reviewer 2 Report

Since the comments have been taken into full consideration, the revised paper could be published after minor revision for the language.

Minor editing of English language required. For example, the title "3. The Proposed System Based Quadrature Multiplexing" might be replaced with "3. The Proposed System Based on  Quadrature Multiplexing".

Author Response

Dear respected referee,

My other authors and I have responded your valuable comment as follows.

Point 1: Minor editing of English language required. For example, the title "3. The Proposed System Based Quadrature Multiplexing" might be replaced with "3. The Proposed System Based on  Quadrature Multiplexing".

Response 1: We have submitted our manuscript for language editing to improve our writing in accordance with the given comment. We have attached the English Editing Certificate in this portal.  "Please see the attachment."

Again, we would like to thank you for your comment.
